# Vitamin C from Seaweed: A Review Assessing Seaweed as Contributor to Daily Intake

**DOI:** 10.3390/foods10010198

**Published:** 2021-01-19

**Authors:** Cecilie Wirenfeldt Nielsen, Turid Rustad, Susan Løvstad Holdt

**Affiliations:** 1Department of Biotechnology and Food Science, Norwegian University of Science and Technology, Sem Sælandsvei 6/8, 7491 Trondheim, Norway; turid.rustad@ntnu.no; 2National Food Institute, Technical University of Denmark, Kemitorvet, 2800 Kongens Lyngby, Denmark; suho@food.dtu.dk

**Keywords:** ascorbic acid, macroalgae, comparison, food, quality, consumption, processing, recommended nutrient intake, dietary reference intake, seasonal variation, analyses, taxonomy

## Abstract

Seaweeds are indiscriminately said to contain significant amounts of vitamin C, but seaweeds are a diverse group, which may limit the ability to generalize. Several studies have been performed on vitamin C in seaweed, and this review covers these findings, and concludes on how much vitamin C is found in seaweeds. A systematic review of vitamin C in 92 seaweed species was conducted followed by analyzing the 132 data entries. The average vitamin C content was 0.773 mg g^−1^ seaweed in dry weight with a 90th percentile of 2.06 mg g^−1^ dry weight. The vitamin C content was evaluated based on taxonomical categories of green, brown and red seaweeds (Chlorophyta (phylum), Phaeophyceae (class), and Rhodophyta (phylum)), and no significant differences were found between them. The vitamin C content was compared to other food sources, and this showed that seaweeds can contribute to the daily vitamin C intake, but are not a rich source. Moreover, seasonal variations, analytical methods, and processing impacts were also evaluated.

## 1. Introduction

Humans are unable to synthesize vitamin C (chemically: ascorbic acid, ascorbate). Humans rely on an adequate supply of vitamin C from their diet and it is therefore considered an essential micronutrient [1]. Vitamin C is fully absorbed and distributed in the human body, with the highest concentrations found in the brain, eye, and adrenal gland [2]. It is involved in collagen synthesis, iron metabolism, tissue growth, and vascular functions, as well as biosynthesis of carnitine, and antioxidant reactions such and inhibiting lipid peroxidation [1,2,3]. Vitamin C is a reductant, meaning it functions as an electron donor and when donating two electrons, it oxidizes into dehydroascorbic acid [1,4]. The vitamin is known for the prevention of scurvy, but may also be able to prevent cardiovascular diseases and some cancer forms [1,2,5]. The Recommended Nutrient Intake (RNI) established by FAO/WHO for vitamin C is 45 mg day^−1^ for adults, which is the amount required to half saturate the body [1]. The Recommended Dietary Allowance (RDA) or Population Reference Intake (PRI) is defined as the average daily level sufficient to meet the nutrient requirements in nearly all healthy individuals. The RDAs established by the Institute for Medicine (U.S.) are 75 and 90 mg day^−1^ for women and men, respectively [6], whereas the PRIs set by EFSA are 95 and 110 mg day^−1^ for women and men, respectively [7]. The Institute for Medicine (2000) describes that smokers dispose of lower vitamin C, even when on a vitamin C rich diet, therefore it is recommended in the United States that the RDA for smokers is increased by 35 mg day^−1^ [6,8].

The Tolerable Upper Intake Level (UL) is not established by the European Food Safety Authorities (EFSA) [9], although the National Institute of Health (USA, NIH) has established a UL for adults of 2000 mg day^−1^ [8], as intakes in that ratio have been shown to produce unpleasant diarrhea and abdominal cramps [1,6,8].

Oxidation of ascorbic acid into dehydroascorbic acid occurs easily due to exposure to high pH, high temperatures, light, oxygen, enzymes as well as exposure to the metals Fe^3+^, Ag^+^, and Cu^2+^ [3]. As dehydroascorbic acid also has corresponding biological importance as ascorbic acid it is important to include both compounds in the analysis of the total vitamin C content of the food [3].

Seaweeds are a common part of the diet in some Asian countries, but they are not traditional in the Western diet. There is an increasing commercial demand for seaweed products due to consumer focus on health and functional foods [10,11]. Some seaweeds contain several ingredients and bioactive compounds, that are of commercial interest such as hydrocolloids, minerals, and polyphenols [12,13].

Several peer-reviewed papers state that seaweeds are also a rich source for vitamin C [4,10,14,15,16] and vitamin C content has been analyzed for various species. More specifically stating that the highest content is found in brown and green algae [17,18] with concentrations of 0.5–3.0 mg g^−1^ dry weight (dw) and red algae containing between 0.1 and 0.8 mg g^−1^ dw [15]. Moreover, Munda (1987) stated that some species have sufficient amounts to cover the recommended daily intake for adults [4]. The statements that seaweeds do in fact contain a significant amount of vitamin C is interesting to investigate. First of all, the term “seaweeds” is very broad and diverse. It is not straight forward to conclude on such a broad term, when it consists of more than 10,000 species divided into green, red, and brown seaweed [19]. The chemical composition of seaweeds are therefore very different in regard to carbohydrates for cell walls and storage, amino acid profile, minerals and pigments, and most likely also when it comes to vitamins [13,20,21]. Secondly, to the best of the authors’ knowledge, an overall picture of the vitamin C content in seaweeds has not yet been established in the literature. It is important to clarify which seaweeds, if any, are rich in vitamin C. A method to establish whether seaweed does in fact contain high contents of vitamin C is to compare the content to the dietary reference intakes such as RNI and other food sources. Examples of foods considered rich in vitamin C are citrus fruits, guava, kiwi, mango, and some berry types [1,22].

The review aims to create a collection of peer-reviewed studies and clarify the vitamin C content of various seaweed species. The data entries from the collection are assessed, evaluated by statistical analysis, and compared with metadata. The influence of processing on the vitamin C content is highlighted, and the method of analysis will be discussed in brief. Lastly, concluding whether seaweed can contribute as a vitamin C source at all or is a rich source of vitamin C compared to other foods.

## 2. Materials and Methods

The work is a systematic review with a meta-analysis performed on the data collected. A thorough search of several studies related to vitamin C in seaweed were identified and collected. Statistical techniques were applied to the data collected to examine and interpret the pooled data to understand the general picture of the vitamin C content in seaweed.

### 2.1. Literature Search

Relevant literature was collected in the period of August to November 2020 from the following databases; Scopus, Web of Science, and the internal university library database DTU Findit. The following keywords and combinations thereof were used: “Vitamin C OR Ascorbic acid”, “Vitamin C content OR Ascorbic acid content” and “seaweed OR macroalgae”. The titles and abstracts were assessed individually for their relevance. If the literature was not available online, the local university library requested and collected scans of the papers.

The sources were collected in Mendeley (Elsevier) and duplicates were removed. The initial criteria for inclusion were; peer-reviewed journals, books, or reports written in English, Norwegian, Danish, or German language. All sources but one (Norwegian) were written in English. Moreover, a criterion for inclusion was that the unit of vitamin C should be expressed in dry weight (dw). Although, in the case where the unit was given in wet weight (ww) or 100 g edible portion, the water content should also be given or achieved from contacting the authors, so a calculation to dry weight was possible.

As the taxonomical names of the seaweed species are updated regularly, the species names were updated to the current official name found in AlgaeBase [19]. The update of the names can be found in the Appendix A.

### 2.2. Data Collection and Meta-Analysis

For the literature review of vitamin C, 34 studies on seaweed were found relevant with a total of 132 inputs. The means and standard deviations from the papers were collected. Some metadata categories were chosen for the review tables; taxonomical order, harvest method, collection site, season, year of harvest, sample treatment, replicates (*n*), and analytical method. In the situations where the research focus of the paper was to study the effect of season or processing, the means from their analyses were kept apart and entered individually. These metadata categories are provided, so an individual assessment of the relevance and reliability of each data entry can be assessed by the reader. In the case of missing metadata, the study was still included in the review. If the unit was not indicated for vitamin C content, the study was excluded from the review. In the analysis of taxonomical categories as well as the comparison to other food sources, all data were included. It is important to be critical to this approach, as some processing might have influenced the vitamin C content, but it was not possible to make a valid objective decision of excluding specific data points, so all data with the correct unit was included. Even though the replicates were given from some of the different studies, each entry in the review tables weighted *n* = 1, when evaluating the data.

It was not possible to conclude on each individual species, as only one or few entries from each species were found in the literature. Therefore, the species were divided into taxonomical categories of green (G), brown (B), and red (R) seaweeds (Chlorophyta (phylum), Phaeophyceae (class), and Rhodophyta (phylum)) as well as taxonomical orders. This was to be able to make a representative overview of their specific vitamin C content.

For data analysis, visualization, and statistics the software program R [23] was used. Statistical analyses were boxplots with mean, median, standard deviations, minimum, and maximum. Moreover in the cases where statistically significant differences were interesting a one-way ANOVA with a Tukey post hoc test was applied. Means were considered significantly different when *p* < 0.05.

## 3. Results and Discussion

### 3.1. Taxonomical Analysis

Vitamin C content is shown in Table 1, Table 2 and Table 3 for a total of 92 species (Phaeophyceae; 36, Rhodophyta; 33 and Chlorophyta; 23). The vitamin C content is given in mg g^−1^ dw. This unit was chosen, as the sample treatment before analysis were different among the papers.

The average content of vitamin C in seaweed from the reviewed studies is 0.773 mg g^−1^ dw. Boxplots for each taxonomical category are seen in Figure 1. The mean for each category is Chlorophyta; 0.781, Phaeophyceae; 0.815, and Rhodophyta; 0.720 mg g^−1^ dw. The range, mean and median of the three categories are not varying considerably, thus no significant differences were found between the categories (one-way ANOVA; *p* = 0.882, F = 0.126). These results are not taking any of the metadata into consideration. The ranges found in this review for each category were broad, and for Rhodophyta the maximum content found was 5.01 mg g^−1^ dw.

The five species with the highest content of vitamin C (above 3.00 mg g^−1^ dw) were *Hydropuntia edulis* (R) > *Dictyota dichotoma* (B) > *Ceramium ciliatum* (R) > *Mesogloia vermiculata* (B) > *Ulva flexuosa* (G) and the 90th percentile of the data entries contained 2.06 mg g^−1^ dw. Their content is comparable to the amount found in peas. Common for the five species is that only one study is published for each of the species, meaning their reliability is not powerful. Data for each individual species are scarce, which is why the seaweed species were divided into the presented categories of green, brown, and red seaweeds. In addition, the entries showed a large variation among species and therefore a broad picture. Looking into the taxonomical order instead of the species, a more reliable and specific estimate of vitamin C content can be achieved.

In Figure 2, Figure 3 and Figure 4, a boxplot for each represented order is shown. No significant differences were found between the orders of Chlorophyta. It can however be seen that seaweeds within the order *Ulvales* (G) have a wide range reaching up to 3.00 mg g^−1^ dw. This indicates that *Ulvales* are richer in vitamin C compared to other green seaweeds. For the orders within the class Phaeophyceae, a statistically significant difference was found (one-way ANOVA; *p* = 0.005, F = 4.334) with a Tukey post-hoc test showing the differences. The statistical results can be found in Figure 3. The order *Ectocarpales* (B) had a high mean of 2.54 mg g^−1^ dw, but all seaweeds within this order was from the same study. To confirm if *Ectocarpales* are high in vitamin C, other studies should look into species within this order. The orders *Fucales* (B) and *Laminariales* (B) have the lowest content of vitamin C within the Phaeophyceae. This is interesting as some of these brown species (*Alaria esculenta, Ascophyllum nodosum*, *Fucus vesiculosus*, *Laminaria* spp., *Saccharina* spp., *Sargassum* spp., and *Undaria pinnatifida*) are of commercial interest [10,11,37,55,56,57,58] and probably useful to avoid claiming they are rich in vitamin C. For Rhodophyta no significance was found between orders, although a broad range was seen for the *Ceramiales* (R). It is worth mentioning that the meta-analysis only considers the taxonomical orders, all other metadata that can influence the vitamin C content such as processing are not included in this analysis.

### 3.2. Comparison to other Foods and RNI

Vitamin C is known to be abundant in rose hips, black and red currants, strawberries, parsley, oranges, and grapefruit [1,59]. In Table 4 the content of vitamin C can be seen for various terrestrial fruits regarded as foods, and the content of seaweeds found in this review. The amount needed to reach the recommended nutrient intake (RNI) is also given as a method to compare the foods and seaweeds. The dietary reference intake RNI is chosen to compare to, as it is an established value set by FAO/WHO on a global evaluation. Other types of dietary reference intakes exist, and those set by EFSA and the Institute of Medicine are all higher and gender-based. This means, that more food production is needed to reach the levels. It is understood by the authors that the RNI is created to consider the entire diet, but it is simply used for comparison between foods.

Rosehip has a high vitamin C content compared to other foods, and to meet the RNI, less than 6 g of rosehip is needed, whereas for seaweed about 400 g is needed based on wet weight. This is half the amount compared to iceberg lettuce, which shows that seaweeds are a better source of vitamin C compared to iceberg lettuce. Although species within the division Rhodophyta and the orders of *Fucales* (B) and *Laminariales* (B) on average contain less than the average of all seaweed species and thereby more than 400 g ww is needed to be consumed to achieve the RNI. It was mentioned that consuming 2–3 g day^−1^ of vitamin C can cause diarrhea [1], but to reach that more than 5 kg ww seaweed should be consumed. It can be concluded based on the reviewed literature that in general seaweeds are not an abundant source of vitamin C for food consumption. Although some can contribute to the daily intake and assist to achieve the RNI, whereas others only have a minimal contribution. Moreover, these results also indicate, that stating either that seaweeds have a fairly high content of vitamin C, or that they have a low content is difficult. “Seaweed” is undiscriminating, and is a category of a large variety of macroalgal species, but the species variation can be significant and therefore conclusions should be made on the level of e.g., taxonomical divisions, order, or species and not on “seaweed”.

### 3.3. Seasonal Variation

Three studies looked into the seasonal variation of vitamin C for the species of brown; *Saccharina latissima*, *Fucus vesiculosus*, and green; *Ulva intestinalis* and *Enteromorpha* spp. [4,30,40,52]. They all found that the highest content of vitamin C was around April–May, with all seaweeds collected in the Northern Hemisphere. It points towards those seasonal fluctuations of vitamin C that occur in seaweed species. Škrovánková (2011) suggests that seaweeds growing closer to the water surface level will contain higher levels of vitamin C than seaweeds harvested from deep waters [17]. This may be due to the higher antioxidant level needed for the seaweed when exposed to high levels of sun, which fits the results seen for seasonal variation.

### 3.4. Analytical Method for Vitamin C

Various analytical methods exist to analyze the vitamin C content in food. The reviewed studies can be divided into three categories. Spectrophotometric methods by reducing cupric ions [4,26], titration with 2,6-dichlorophenolindophenol (Titration) [18,24,36,40,43,45,47,48,51], and chromatography such as High Performance Liquid Chromatography (HPLC) [25,29,30,31,32,33,35,38,39,41,42,44,46,49,50,52,53]. Five papers did not mention the analytical method used [27,28,34,37,54].

It is worth to mention that indophenol titration is the official AOAC Method 967.21 for juices [60,61], although many studies on vitamin C in foodstuff are performed by chromatographic methods [3], which is the AOAC First Action Official Method 2012.22 for infant formula and nutritional formulas [62]. No studies referred to the AOAC 984.26-1985, Vitamin C (Total) in Food-Semiautomated Fluor [63].

Quantification of vitamin C by different analytical methods with the same samples was not performed by any study, and it was therefore not possible to conclude the effect of the analytical method. In the case where two or more studies had analyzed the same species with different analytical methods, no specific trend was found such as one analysis always quantifying a higher content. The results, therefore, indicate that even though one analysis might be over- or underestimating, other factors such as biological variations, season, harvest site, sample treatment or other unknown factors can influence the result as well.

### 3.5. Processing and the Influence on Vitamin C

Vitamin C is somewhat easily degraded, and in nutrient stability studies in foods, it is assumed that if vitamin C is well retained, then other nutrients will be just as well retained if not better. The degradation of vitamin C depends on moisture-, oxygen, light, and metal ion catalysis as well as temperature and pH [5,64]. An analysis of the metadata of processing was performed in this present review (data not shown). No trend for washing methods or drying methods was observed. Therefore it was not possible to conclude the effect of processing on vitamin C in seaweed based on the overall reviewed literature.

A few studies looked into the degradation of vitamin C [18,39,40]. Both Sappati et al. (2019) and Chan et al. (1997) found that sun drying and oven drying at temperatures between 30 and 70 °C had a significant, negative influence on the vitamin C content in *Saccharina latissima* (B) and *Sargassum hemiphyllum* (B), respectively. Vegetable blanching and boiling can be performed to reduce microbial load and inactivate enzymes, but it is known that it also compromises quality compounds such as vitamin C [65,66]. Amorim et al. (2012) studied the influence of 20 min. boiling on *Undaria pinnatifida* (B) and found the reduction of vitamin C to be below the detection limit [39]. Amorim-Carrilho et al. (2014) studied different processing methods on *Himanthalia elongata* (B). They found that 15 min boiling in 100 °C water, rehydration in water for 10 min, and steaming for 40 min reduced the vitamin C content below the detectable limit [32].

Friedlander (1989) found that seven months of storage decreased the initial value of ascorbic acid in *Pyrophyllon subtumens* (R) and *Pyropia columbina* (R) to 15% and 34%, respectively [42]. Some vegetables can lose up to 70% of the initial content during storage [19]. Balan et al. (2016) suggested that a matrix with a fibrous texture and low water content would preserve ascorbic acid better during storage [66], which could be an interesting hypothesis for dried seaweed. Friedlander (1989) also found that drying at 30 °C for 4 h did not affect the ascorbic acid content, moreover did washing or toasting for 15 s of nori sheets not influence the ascorbic acid content.

Although no studies looked into the effect of cutting the seaweed biomass, a study on rose hips showed that cutting would lead to a decrease in vitamin C content, which might also be the case for seaweeds [64].

## 4. Conclusions

Seaweeds are not a rich source of vitamin C, but when consumed they feed into the daily intake. To reach the Recommended Nutrient Intake approximately 400 g ww of seaweed should be consumed per day, which in contrast to rosehip is 5.35 g ww.

The vitamin C content can vary, due to biological, seasonal, locational, and treatment variations. Moreover, evaluating and generalizing seaweeds can be difficult, the nutritional quality should be evaluated based on e.g., taxonomical category, order, or species. The mean content in seaweed is 0.773 mg g^−1^ dw with a 90 percentile of 2.06 mg g^−1^ dw. A study of the taxonomical orders of the species indicated that the green seaweeds *Ulvales* contained up to 3.00 mg g^−1^. Whereas, brown species within the orders *Fucales* and *Laminarales* had low amounts of vitamin C.

It was found that drying, boiling and long storage time lead to a decrease in vitamin C in seaweed, as it is easily oxidized.

## Figures and Tables

**Figure 1 foods-10-00198-f001:**
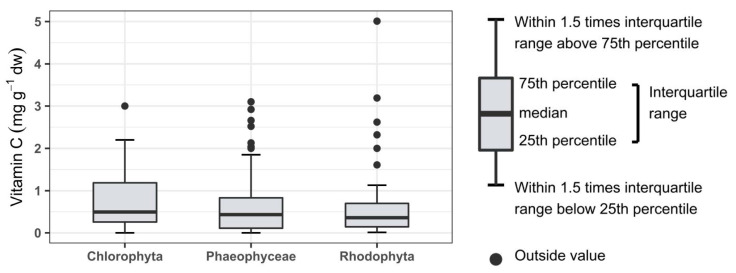
Data analysis of vitamin C content (mg g^−1^ dw) represented in boxplots and statistical output for the three categories; Chlorophyta (phylum), Phaeophyceae (class), and Rhodophyta (phylum).

**Figure 2 foods-10-00198-f002:**
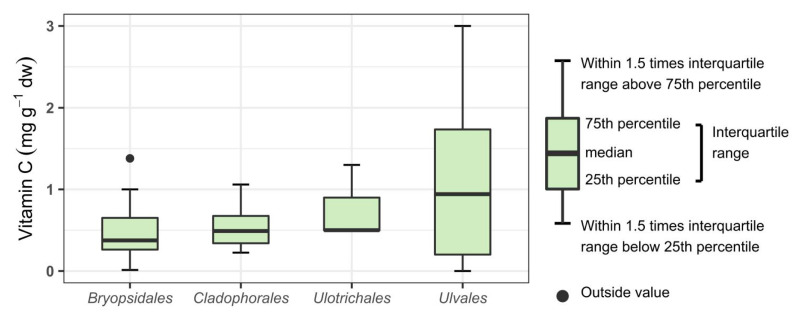
Data analysis of vitamin C content (mg g^−1^ dw) represented in boxplots and statistical output for some orders of the phylum Chlorophyta.

**Figure 3 foods-10-00198-f003:**
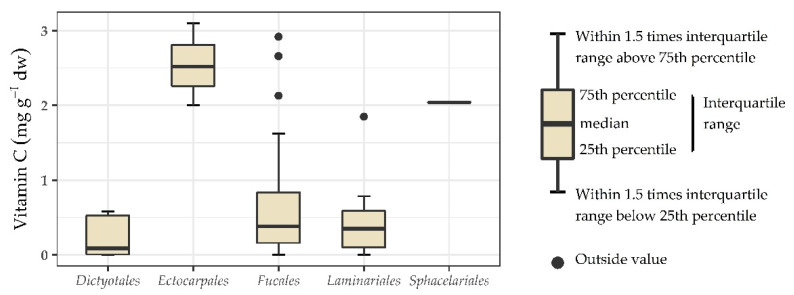
Data analysis of vitamin C (mg g^−1^ dw) content represented in boxplots and statistical output for some orders of the class Phaeophyceae. The letters “a” and “b” indicate statistically significant differences between orders.

**Figure 4 foods-10-00198-f004:**
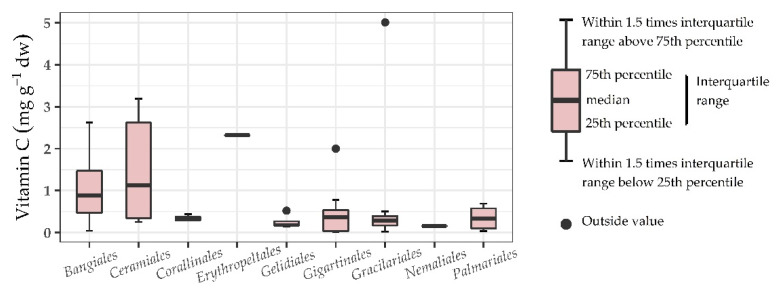
Data analysis of vitamin C content (mg g^−1^ dw) represented in boxplots and statistical output for some orders of the phylum Rhodophyta.

**Table 1 foods-10-00198-t001:** Vitamin C in Phaeophyceae (Class). Species arranged by orders.

Species	Origin				Sample Treatment	*n*	Analytical Method	Vitamin C mg g^−1^ * ± SD	Reference
	Wild/ Cultivated	Collection Site	Season	Year					
**Dictyotales**									
*Dictyota dichotoma*	Wild	Piran, Slovenia	January–November	1984	Vacuum-dried at 30 °C	8	Spectrophoto-metrically	3.79 ± 0.44	[4]
*Padina gymnospora*	Wild	Tanjung Tuan, Malaysia	-	2008 ***	Washed in running water	-	Titration	0.085	[24]
*Padina gymnospora*	Wild	Hurghada, Egypt	April–June	2019 ***	Washed with tap and distilled water, air-dried	1	Chromatography	0.006	[25]
*Padina pavonica*	Wild	Piran, Slovenia	October–November	1984	Vacuum-dried at 30 °C	2	Spectrophoto-metrically	0.58 ± 0.26	[4]
*Padina tetrastromatica*	Wild	Visakhapatnam, India	Yearly	1996–1997	Washed in fresh water and freeze-dried	12	Spectrophoto-metrically	0.525	[26]
**Ectocarpales**									
*Ectocarpus siliculosus*	Wild	Piran, Slovenia	January–June	1984	Vacuum-dried at 30 °C	5	Spectrophoto-metrically	2.52 ± 0.58	[4]
*Mesogloia vermiculata*	Wild	Piran, Slovenia	April	1984	Vacuum-dried at 30 °C	1	Spectrophoto-metrically	3.10	[4]
*Scytosiphon lomentaria*	Wild	Piran, Slovenia	January–May	1984	Vacuum-dried at 30 °C	4	Spectrophoto-metrically	2.00 ± 0.26	[4]
**Fucales**									
*Ascophyllum nodosum*	Wild	-	-	1920	-	-	-	0.55–1.65	[27]
*Ascophyllum nodosum*	-	-	-	-	-	-	-	0.082	[28]
*Carpodesmia crinita*	Wild	Piran, Slovenia	March–November	1984	Vacuum-dried at 30 °C	5	Spectrophoto-metrically	1.62 ± 0.28	[4]
*Cystoseira compressa*	Wild	Piran, Slovenia	January–November	1984	Vacuum-dried at 30 °C	9	Spectrophoto-metrically	2.13 ± 0.28	[4]
*Durvillaea antarctica*	Wild	Santa Ana, Chile	October–December	2012	Washed in deionized water and dried at 20 °C	3	Chromatography	0.348	[29]
*Fucus vesiculosus*	Wild	Chincoteague Island, USA	May	-	Freeze-dried	-	Chromatography	0.517 ± 0.078	[30]
*Fucus vesiculosus*	Wild	Chincoteague Island, USA	August	-	Freeze-dried	-	Chromatography	0.409 ± 0.101	[30]
*Fucus virsoides*	Wild	Piran, Slovenia	January–November	1984	Vacuum-dried at 30 °C	9	Spectrophoto-metrically	2.66 ± 0.45	[4]
*Himanthalia elongata*	Wild	Galicia, Spain	-	2010	Fresh	2	Chromatography	2.92 ± 0.37 **	[31]
*Himanthalia elongata*	Bought	Galicia, Spain	-	2014 ***	Dehydrated	6	Chromatography	0.207 ± 0.09	[32]
*Himanthalia elongata*	Wild	Galicia, Spain	December	2015	Dried < 38 °C	3	Chromatography	0.692 ± 0.053	[33]
*Polycladia myrica*	Wild	Hurghada, Egypt	April–June	2019 ***	Washed with tap and distilled water, air-dried	1	Chromatography	0.008	[25]
*Sargassum baccularia*	Wild	Tanjung Tuan, Malaysia	-	2008 ***	Washed in running water	-	Titration	0.224	[24]
*Sargassum cervicorne*	Wild	Tanjung Tuan, Malaysia	-	2008 ***	Washed in running water	-	Titration	0.254	[24]
*Sargassum hemiphyllum*	Wild	Tung Ping Chau, Hong Kong	December	1995	Washed then sun-dried for 4 days	3	Titration	0.519 ± 0.035	[18]
*Sargassum hemiphyllum*	Wild	Tung Ping Chau, Hong Kong	December	1995	Washed then oven dried for 15 h at 60 °C	3	Titration	0.977 ± 0.121	[18]
*Sargassum hemiphyllum*	Wild	Tung Ping Chau, Hong Kong	December	1995	Washed then freeze-dried	3	Titration	1.53 ± 0.12	[18]
*Sargassum latifolium*	Wild	Hurghada, Egypt	April–June	2019 ***	Washed with tap and distilled water, air-dried	1	Chromatography	0.007	[25]
*Sargassum mcclurei*	Wild	Nha Trang, Vietnam	June	2003	-	1	-	0.657 **	[34]
*Sargassum muticum*	Wild	Bourgneuf Bay, France	-	-	Fresh	-	Chromatography	0.560	[35]
*Sargassum muticum*	Wild	Hurghada, Egypt	April–June	2019 ***	Washed with tap and distilled water, air-dried	1	Chromatography	0.012	[25]
*Sargassum polycystum*	-	Kota Kinabalu, Malaysia	-	2009 ***	Washed with distilled water	3	Titration	0.383 ± 0.000 **	[36]
*Sargassum* spp.	Wild	Hurghada, Egypt	April–June	2019 ***	Washed with tap and distilled water, air-dried	1	Chromatography	0.004	[25]
*Sargassum tenerrimum*	Wild	Visakhapatnam, India	Yearly	1996–1997	Washed in fresh water and freeze-dried	11	Spectrophoto-metrically	0.280	[26]
*Sargassum vulgare*	Wild	Visakhapatnam, India	Yearly	1996–1997	Washed in fresh water and freeze-dried	1	Spectrophoto-metrically	0.300	[26]
*Turbinaria conoides*	Wild	Tanjung Tuan, Malaysia	-	2008 ***	Washed in running water	-	Titration	0.112	[24]
*Turbinaria* spp.	Wild	Hurghada, Egypt	April–June	2019 ***	Washed with tap and distilled water, air-dried	1	Chromatography	0.008	[25]
**Laminariales**									
*Alaria* spp.	-	-	-	-	-	-	-	0.0221–0.497	[37]
*Eisenia arborea*	Wild	Bahía Asunción, Mexico	March–December	-	Sun-dried	10	Chromatography	0.344 ± 0.06	[38]
*Laminaria digitata*	-	-	-	-	-	-	-	0.355	[28]
*Laminaria ochroleuca*	Wild	Galicia, Spain	December	2015	Dried < 38 °C	3	Chromatography	0.785 ± 0.092	[33]
*Laminaria* spp.	Wild	Redondela, Spain	February	2011	Fresh	6	Chromatography	nd	[39]
*Laminaria* spp.	Wild	Galicia, Spain	-	2010	Fresh	2	Chromatography	0.096 ± 0.004 **	[31]
*Saccharina latissima*	Cultivated	Damariscotta Bay, USA	Early May	2017	Washed in running water	3	Titration	0.611 ± 0.074	[40]
*Saccharina latissima*	Cultivated	Damariscotta Bay, USA	Late June	2017	Washed in running water	3	Titration	0.452 ± 0.009	[40]
*Saccharina latissima*	Cultivated	Damariscotta Bay, USA	May and June	2017	Washed in running water and dried in various ways	16	Titration	0.104 ± 0.016 **	[40]
*Undaria pinnatifida*	Wild	Redondela, Spain	February	2011	Fresh	6	Chromatography	0.118 ± 0.022	[39]
*Undaria pinnatifida*	Wild	Redondela, Spain	February	2011	Boiling 20 min.	6	Chromatography	nd	[39]
*Undaria pinnatifida*	-	-	-	-	-	-	-	1.85	[28]
*Undaria pinnatifida*	Wild	Galicia, Spain	December	2015	Dried < 38 °C	3	Chromatography	0.693 ± 0.090	[33]
Wakame	Bought	-	-	2008 ***	-	-	Titration	0.030	[24]
**Sphacelariales**									
*Halopteris scopari*	Wild	Piran, Slovenia	January–November	1984	Vacuum-dried at 30 °C	9	Spectrophoto-metrically	2.04 ± 0.48	[4]

* mg ascorbic acid per g dry weight; ** mean of given numbers on various drying methods with no significant differences; *** year of publication, year of harvest not given; nd, not detected

**Table 2 foods-10-00198-t002:** Vitamin C in Rhodophyta (phylum). Species arranged by orders.

Species	Origin				Sample Treatment	*n*	Analytical Method	Vitamin C mg g^−1^ * ± SD	Reference
	Wild/ Cultivated	Collection Site	Season	Year					
**Bangiales**									
Nori	Bought	-	-	2008 ***	Washed in running water, freeze-dried	-	Titration	0.390	[24]
*Pyropia acanthophora*	Wild	Central West Coast, India	July	2013	Washed in seawater, shade dried	5	Chromatography	0.042 ± 0.019	[41]
*Pyropia columbina*	Wild	Brighton, New Zealand and Dunedin, New Zealand	June–October	1986	Washed with seawater and oven-dried at 30 °C	7	Chromatography	2.62 ± 0.68	[42]
*Porphyra* spp.	Wild	Galicia, Spain	December	2014	Dried < 38 °C	3	Chromatography	0.712 ± 0.102	[33]
*Porphyra umbilicalis*	-	-	-	-	-	-	-	1.61	[28]
*Porphyra umbilicalis*	Wild	Galicia, Spain	-	2010	Fresh	2	Chromatography	1.05 ± 0.27 **	[31]
**Ceramiales**									
*Centroceras clavulatum*	Wild	Visakhapatnam, India	Yearly	1996–1997	Washed in fresh water and freeze-dried	1	Spectrophoto-metrically	0.345	[26]
*Ceramium ciliatum*	Wild	Piran, Slovenia	March–June	1984	Vacuum-dried at 30 °C	4	Spectrophoto-metrically	3.19 ± 0.51	[4]
*Halopithys incurva*	Wild	Piran, Slovenia	January–November	1984	Vacuum-dried at 30 °C	9	Spectrophoto-metrically	1.13 ± 0.24	[4]
*Laurencia obtusa*	Wild	Khanh Hoa, Vietnam	July	2003	-	1	-	0.252**	[34]
*Nitophyllum punctatum*	Wild	Piran, Slovenia	April	1984	Vacuum-dried at 30 °C	1	Spectrophoto-metrically	2.62	[4]
**Corallinales**									
*Amphiroa fragilissima*	Wild	Visakhapatnam, India	Yearly	1996–1997	Washed in fresh water and freeze-dried	11	Spectrophoto-metrically	0.285	[26]
*Jania rubens*	Wild	Piran, Slovenia	July	1984	Vacuum-dried at 30 °C	1	Spectrophoto-metrically	0.436	[4]
*Jania rubens*	Wild	Visakhapatnam, India	Yearly	1996–1997	Washed in fresh water and freeze-dried	1	Spectrophoto-metrically	0.310	[26]
**Erythropeltales**									
*Pyrophyllon subtumens*	Wild	Brighton, New Zealand	June–October	1986	Washed with seawater and oven-dried at 30 °C	4	Chromatography	2.32 ± 0.33	[42]
**Gelidiales**									
*Gelidiella acerosa*	Wild	Khanh Hoa, Vietnam	July	2003	-	1	-	0.522 **	[34]
*Gelidium pusillum*	Wild	Visakhapatnam, India	Yearly	1996–1997	Washed in fresh water and freeze-dried	2	Spectrophoto-metrically	0.150	[26]
*Millerella myrioclada*	Wild	Visakhapatnam, India	Yearly	1996–1997	Washed in fresh water and freeze-dried	1	Spectrophoto-metrically	0.185	[26]
*Pterocladia heteroplatos*	Wild	Visakhapatnam, India	Yearly	1996–1997	Washed in fresh water and freeze-dried	11	Spectrophoto-metrically	0.175	[26]
**Gigartinales**									
*Callophyllis variegata*	Wild	Santa Ana, Chile	October–December	2012	Washed in deionized water and dried at 20 °C	3	Chromatography	0.011	[29]
*Chondrus crispus*	Wild	Galicia, Spain	December	2014	Dried < 38 °C	3	Chromatography	0.538 ± 0.055	[33]
*Eucheuma denticulatum*	Cultivated	Sulawesi, Indonesia	February	2016	Washed with distilled water and dried	2	Titration	0.035 ± 0.006	[43]
*Eucheuma denticulatum*	Wild	O’ahu, USA	February	2002	Washed in filtered seawater and dried at 60 °C in an air oven	1	Chromatography	2.0	[44]
*Hypnea musciformis*	Wild	Visakhapatnam, India	Yearly	1996–1997	Washed in fresh water and freeze-dried	9	Spectrophoto-metrically	0.370	[26]
*Hypnea valentiae*	Wild	Nha Trang, Vietnam	July	2003	-	1	-	0.438 **	[34]
*Kappaphycus alvarezii*	Cultivated	Bangi Sabah, Malaysia	-	2009 ***	Washed with distilled water	3	Titration	0.395 ± 0.000 **	[36]
*Kappaphycus alvarezii*	Cultivated	Popayato, Indonesia	-	2020 ***	Washed in distilled water and dried	2	Titration	0.033 ± 0.001	[45]
*Kappaphycus alvarezii*	Wild	Khanh Hoa, Vietnam	May	2003	-	1	-	0.551 **	[34]
*Kappaphycus alvarezii*	Cultivated	Sulawesi, Indonesia	February	2016	Washed with distilled water and dried	2	Titration	0.036 ± 0.006	[43]
*Kappaphycus alvarezii*	-	West Coast, India	-	2005	Dried for 6 h at 50 °C	3	Chromatography	0.107 ± 0.30	[46]
*Kappaphycus striatum*	Cultivated	Sulawesi, Indonesia	February	2016	Washed with distilled water and dried	2	Titration	0.035 ± 0.006	[43]
*Sphaerococcus coronopifolius*	Wild	Marmara, Turkey	June	2009	Washed in tap water and dried at room temperature	4	Titration	0.78 ± 0.07	[47]
**Gracilariales**									
*Crassiphycus changii*	Wild	Tanjung Tuan, Malaysia	-	2008 ***	Washed in running water	-	Titration	0.285	[24]
*Crassiphycus changii*	Cultivated	Kedah, Malaysia	-	2000 ***	Washed in running water	3	Titration	0.285 **	[48]
*Crassiphycus changii*	Wild	Santubong, Malaysia	-	2017 ***	Washed with distilled water and freeze-dried	3	Chromatography	0.025 ± 0.002	[49]
*Gracilaria corticata*	Wild	Visakhapatnam, India	Yearly	1996–1997	Washed in fresh water and freeze-dried	12	Spectrophoto-metrically	0.100	[26]
*Gracilaria gracilis*	Wild	Marmara, Turkey	June	2009	Washed in tap water and dried at room temperature	4	Titration	0.24 ± 0.01	[47]
*Gracilaria tenuistipitata*	Wild	Nha Trang, Vietnam	May	2003	-	1	-	0.502 **	[34]
*Hydropuntia edulis*	Wild	Thondi, India	-	2015 ***	Washed in fresh water, shade dried 28 °C	2	Chromatography	5.01 ± 0.40	[50]
**Nemaliales**									
*Liagora albicans*	Wild	Visakhapatnam, India	Yearly	1996–1997	Washed in fresh water and freeze-dried	1	Spectrophoto-metrically	0.155	[26]
**Palmariales**									
Dulse	Bought	-	-	2008 ***	Dried	-	Titration	0.120	[24]
*Palmaria palmata*	-	-	-	-	-	-	-	0.69	[28]
*Palmaria palmata*	Wild	Galicia, Spain	-	2010	Fresh	2	Chromatography	0.039 ± 0.001 **	[31]
*Palmaria palmata*	Wild	Bretagne, France	December	2014	Dried < 38 °C	3	Chromatography	0.538 ± 0.055	[33]

* mg ascorbic acid per g dry weight; ** calculated from wet weight to dry weight based on given proximate composition; *** year of publication, year of harvest not given; nd, not detected.

**Table 3 foods-10-00198-t003:** Vitamin C in Chlorophyta (phylum). Species arranged by orders.

Species	Origin				Sample Treatment	*n*	Analytical Method	C Vitamin mg g^−1^ * ± SD	Reference
	Wild/ Cultivated	Collection Site	Season	Year					
**Bryopsidales**									
*Bryopsis pennata*	Wild	Visakhapatnam, India	Yearly	1996–1997	Washed in fresh water and freeze-dried	4	Spectrophoto-metrically	0.250	[26]
*Caulerpa lentillifera*	Wild	Tanjung Tuan, Malaysia	-	2008 ***	Washed in running water	-	Titration	0.274	[24]
*Caulerpa lentillifera*	Wild	Amphor BanLam, Thailand	March	2014 ***	Washed in running water	3	Titration	0.013 **	[51]
*Caulerpa lentillifera*	-	Semporna, Malaysia	-	2009 ***	Washed with distilled water	3	Titration	0.389 ± 0.000 **	[36]
*Caulerpa racemosa*	Wild	Tanjung Tuan, Malaysia	-	2008 ***	Washed in running water	-	Titration	0.225	[24]
*Caulerpa racemosa*	Wild	Visakhapatnam, India	Yearly	1996–1997	Washed in fresh water and freeze-dried	9	Spectrophoto-metrically	0.275	[26]
*Caulerpa racemosa*	Wild	Khanh Hoa, Vietnam	July	2003	-	1	-	0.912 **	[34]
*Caulerpa sertularioides*	Wild	Visakhapatnam, India	Yearly	1996–1997	Washed in fresh water and freeze-dried	4	Spectrophoto-metrically	0.375	[26]
*Caulerpa taxifolia*	Wild	Visakhapatnam, India	Yearly	1996–1997	Washed in fresh water and freeze-dried	7	Spectrophoto-metrically	0.390	[26]
*Codium tomentosum*	Wild	Marmara, Turkey	June	2009	Washed in tap water and dried at room temperature	4	Titration	1.38 ± 0.19	[47]
*Codium vermilara*	Wild	Piran, Slovenia	July	1984	Vacuum-dried at 30 °C	1	Spectrophoto-metrically	1.00	[4]
**Cladophorales**									
*Chaetomorpha antennina*	Wild	Visakhapatnam, India	Yearly	1996–1997	Washed in fresh water and freeze-dried	9	Spectrophoto-metrically	0.490	[26]
*Chaetomorpha brachygona*	Wild	Visakhapatnam, India	Yearly	1996–1997	Washed in fresh water and freeze-dried	8	Spectrophoto-metrically	0.225	[26]
*Cladophora rupestris*	Wild	Piran, Slovenia	November	1984	Vacuum-dried at 30 °C	1	Spectrophoto-metrically	1.06	[4]
*Cladophora socialis*	Wild	Visakhapatnam, India	Yearly	1996–1997	Washed in fresh water and freeze-dried	2	Spectrophoto-metrically	0.340	[26]
*Cladophora* spp.	Wild	Visakhapatnam, India	Yearly	1996–1997	Washed in fresh water and freeze-dried	1	Spectrophoto-metrically	0.675	[26]
**Ulotrichales**									
*Acrosiphonia orientalis*	Wild	Visakhapatnam, India	Yearly	1996–1997	Washed in fresh water and freeze-dried	12	Spectrophoto-metrically	0.500	[26]
*Gayralia oxysperma*	Wild	Hawai’i, USA	October	2001	Washed in filtered seawater and dried at 60 °C in an air oven	1	Chromatography	1.3	[44]
*Monostroma nitidum*	Wild	Nha Trang, Vietnam	May	2003	-	1	-	0.495 **	[34]
**Ulvales**									
*Ulva compressa*	Wild	Visakhapatnam, India	Yearly	1996–1997	Washed in fresh water and freeze-dried	6	Spectrophoto-metrically	0.310	[26]
*Ulva flexuosa*	Wild	O’ahu, USA	January	2002	Washed in filtered seawater and dried at 60 °C in an air oven	1	Chromatography	3.0	[44]
*Ulva intestinalis*	Wild	Muğla, Turkey	August	2013	Washed in fresh water, frozen, thawed, dried at 40 °C for 24 hrs	3	Chromatography	0.028 ± 0.001	[52]
*Ulva intestinalis*	Wild	Muğla, Turkey	November	2013	Washed in fresh water, frozen, thawed, dried at 40 °C for 24 hrs	3	Chromatography	0.034 ± 0.000	[52]
*Ulva intestinalis*	Wild	Muğla, Turkey	January	2014	Washed in fresh water, frozen, thawed, dried at 40 °C for 24 hrs	3	Chromatography	0.026 ± 0.000	[52]
*Ulva intestinalis*	Wild	Muğla, Turkey	April	2014	Washed in fresh water, frozen, thawed, dried at 40 °C for 24 hrs	3	Chromatography	1.47 ± 0.02	[52]
*Ulva lactuca*	Wild	O’ahu, USA	January	2002	Washed in filtered seawater and dried at 60 °C in an air oven	1	Chromatography	2.2	[44]
*Ulva lactuca*	Wild	Visakhapatnam, India	Yearly	1996–1997	Washed in fresh water and freeze-dried	12	Spectrophoto-metrically	0.155	[26]
*Ulva reticulata*	Wild	Nha Trang, Vietnam	March	2003	-	1	-	0.971 **	[34]
*Ulva reticulata*	Wild	Pattani Bay, Thailand	May	2014 ***	Washed in running water	3	Titration	0.00 **	[51]
*Ulva rigida*	Wild	Marmara, Turkey	June	2009	Washed in tap water and dried at room temperature	4	Titration	2.05 ± 0.33	[47]
*Ulva rigida*	Wild	Northern Adriatic	January–November	1984	Vacuum-dried at 30 °C	9	Spectrophoto-metrically	2.00 ± 0.52	[4]
*Ulva rigida*	Wild	Northwest Iberian coast, Spain	-	2010 ***	-	3	Chromatography	0.942	[53]
*Ulva* spp.	Wild	Piran, Slovenia	January–May	1984	Vacuum-dried at 30 °C	4	Spectrophoto-metrically	2.04 ± 0.34	[4]
*Ulva* spp.	Wild	Piran, Slovenia	October–November	1984	Vacuum-dried at 30 °C	2	Spectrophoto-metrically	1.23 ± 0.35	[4]
*Ulva* spp.	Wild	Visakhapatnam, India	Yearly	1996–1997	Washed in fresh water and freeze-dried	1	Spectrophoto-metrically	0.420	[26]
*Ulva* spp.	Wild	Locquirec, France	June–September	1982	Rinsed with seawater	5	-	0.247 ± 0.278 **	[54]
*Ulva* spp.	-	-	-	-	-	-	-	1.25	[28]
*Ulva* spp.	Wild	Galicia, Spain	December	2015	Dried < 38 °C	3	Chromatography	0.746 ± 0.136	[33]

* mg ascorbic acid per g dry weight; ** calculated from wet weight to dry weight based on given proximate composition; *** year of publication, year of harvest not given nd, not detected.

**Table 4 foods-10-00198-t004:** Vitamin C content found in other food sources as well as these reviewed data categorized on different levels of the taxonomy. Moreover, the amount that is assumed necessary to consume to meet the RNI. All the contents from other foods are calculated based on [22]. The list is made in descending order with seaweeds shaded in the color of the seaweed categories.

Food	mg Vitamin C g^−1^ dw	g ww to Meet RNI; 45 mg Day^−1^
Rosehip	36.4	5.35
Parsley	20.8	14.6
Broccoli	10.1	40.1
Black currant	8.66	24.9
Strawberry	6.67	67.5
Grapefruit	4.08	95.1
Ectocarpales *	2.54	118
Peas	2.11	105
90th percentile seaweed *	2.06	146
Potato	1.29	170
Iceberg lettuce	1.17	818
Dictyotales *	0.997	301
Chlorophyta *	0.781	384
Average seaweed *	0.773	388
Rhodophyta *	0.720	417
Fucales *	0.686	437
Laminariales *	0.496	605

* the amount needed of macroalgae (g ww) calculated based on the assumption of a moisture content of 85% ww. Brown, Phaeophyceae; grey, seaweed in general; green, Chlorophyta; red, Rhodophyta.

## Data Availability

All data collected and analyzed behind this review are openly available in FigShare at DOI:10.11583/DTU.13369874.

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
