# Peer review of "Vitamin C from Seaweed: A Review Assessing Seaweed as Contributor to Daily Intake"

_foods, 2021, doi:10.3390/foods10010198_

Round 1

Reviewer 1 Report

Manuscript ID: foods-1058321

Title:  Vitamin C in seaweed: a review and assessment of various species in regards to quality and human consumption

Major comments:

Please revise the title of manuscript as “Vitamin C from seaweed: A review with assessment of various seaweed species for quality and human consumption” OR you can improve it.

Introduction section need significant revision, it is very short and please highlight the knowledge gap. Only few references are used to write the introduction section, please use recent reference and each statement need to cite with appropriate reference.

Please replace all Figure with high quality resolution.  

Please insert each table at appropriate place in the manuscript.  

Conclusion section need to be revised, please provide the brief conclusion at least 10-15 lines with key highlights of this review.

Author Response

pdf is attached

Reviewer 2 Report

I have carefully reviewed manuscript entitled: “Vitamin C in seaweed: a review and assessment of various species in regards to quality and human consumption”.

This manuscript is very interesting and presents important data concerning the content of vitamin C in different species of seaweeds. Usually it is assumed that seaweeds are a rich source of vitamins, especially vitamin C. The Authors presented in an understandable way the current facts about this vitamins in seaweeds.

I have some minor comments on this manuscript, I hope they will help to improve this review.

Comments:
Line 23: I propose “seaweeds” instead of macroalgae. It can be later mentioned that “seaweeds” and “macroalgae” are used interchangeably
Line 35: shouldn’t be “UIL” – please check
Line 38: shouldn’t be Cu (copper) instead of CU?
Line 71: I propose full “dry weight” here, in this line
Line 92: comma after “(phylum)”, not dot
Line 114: In my opinion, legend should be added to this Figure. Gray dot is a mean, I assume that the thick line is a median and the box is upper (3rd, Q3) and lower (1st, Q1) quartile. What about whisker? Please explain for readers who are not familiar with statistics
Line 118: I propose to add to seaweeds names the abbreviations (B), (G), (R), which correspond to brown, green and red seaweeds. You can introduce these abbreviations in Line 92. Then it will be easier to compare seaweeds, which phylum/class can have higher contents of vitamin C. Not all readers know all species of seaweeds but division because of the colors can facilitate analysis of the results. I propose to add these abbreviations to all seaweeds species listed in the text of manuscript.
Line 128: “this indicates…” please check English, there are some minor mistakes
Line 130: “difference was….”
Line 133: “should look….”
Line 135: I propose to list here these species
Line 137: shouldn’t be “worth mentioning…”?
Line 139: shouldn’t be “included in….”
Line 141: in Figures you present here values. I assume it is the value of mean. In the boxplot when you present mean ± SD it is a box and ±1.96 SD is the end of whisker. Here, this value looks for me as ±1.96 SD. If it is the value of mean, it should be close to the gray dot. Please check and explain. The same remark to all Figures.
Line 170: “macroalgae” instead of “macro algae”
Line 178: I propose “…that the highest content of vitamin C was around April-May…” – please check
Line 179: “occur”
Line 216: “Amorim-Carrilho et al.” – please add year
Line 253: please standardize – in Table should be mg g-1. Also in the case of the unit of temperature ºC or ºC. Please take a look also in the text of manuscript
Alaria spp. – spp. without italics
Line 297: Latin names in Italics - Sargassum hemiphyllum, please correct in the whole section
Line 312: small letters in the title of publication, please correct in the whole section

Author Response

pdf is attached

Round 2

Reviewer 1 Report

Authors addressed major comments but figure quality is still worst and not visible. Please replace all figures with high resolution figures.